# RobuRCDet: Enhancing Robustness of Radar-Camera Fusion in Bird's Eye View for 3D Object Detection

**Jingtong Yue**[1,2*]       **Zhiwei Lin**[1*]       **Xin Lin**[2*]       **Xiaoyu Zhou**[1]

**Xiangtai Li**[1]       **Lu Qi**[4]       **Yongtao Wang**[1]       **Ming-Hsuan Yang**[3]

[1]Peking University    [2]Sichuan University    [3]UC Merced    [4]Insta360 Research

## Abstract

While recent low-cost radar-camera approaches have shown promising results in multi-modal 3D object detection, both sensors face challenges from environmental and intrinsic disturbances. Poor lighting or adverse weather conditions degrade camera performance, while radar suffers from noise and positional ambiguity. Achieving robust radar-camera 3D object detection requires consistent performance across varying conditions, a topic that has not yet been fully explored. In this work, we first conduct a systematic analysis of robustness in radar-camera detection on five kinds of noises and propose RobuRCDet, a robust object detection model in bird's eye view (BEV). Specifically, we design a 3D Gaussian Expansion (3DGE) module to mitigate inaccuracies in radar points, including position, Radar Cross-Section (RCS), and velocity. The 3DGE uses RCS and velocity priors to generate a deformable kernel map and variance for kernel size adjustment and value distribution. Additionally, we introduce a weather-adaptive fusion module, which adaptively fuses radar and camera features based on camera signal confidence. Extensive experiments on the popular benchmark, nuScenes, show that our RobuRCDet achieves competitive results in regular and noisy conditions. The source code will be released at https://github.com/Jingtong0527/RobuRCDet.

## 1 Introduction

Multi-modal 3D object detection is crucial in computer vision, as it leverages the complementary signals captured by cameras and 3D sensors. Due to its accurate 3D depth information and robustness, radar has emerged as a promising and cost-effective 3D signal, benefiting applications such as autonomous driving (Qi et al., 2019; Liu et al., 2024) and robotic navigation (Arnold et al., 2019; Wan et al., 2024). Despite the significant success of previous radar-camera 3D object detection methods (Kim et al., 2024; Lin et al., 2024; Zhou et al., 2023; Jiang et al., 2024), they often neglect the importance of model robustness, which limits the practical applicability of these methods.

Several works (Zhou et al., 2023; Kim et al., 2023b; Lin et al., 2024) have designed practical feature encoders and multi-modal fusion modules to enhance model robustness in challenging scenarios, such as fewer input sweeps, single sensor failure, or extended radar perception range. However, they usually neglect the interference caused by adverse weather and lighting conditions on camera signals, the impact of internal and external factors on radar signals, and the cooperative effects between the two sensors, as shown in Figure 1.

This motivates us to systematically analyze various types of noise in radar-camera 3D object detection and then propose a robust method to counteract interference. However, there are very few datasets that encompass all possible scenarios. Take Radiate dataset (Sheeny et al., 2021) as an example; its camera data is limited to two views (left and right) and is mainly provided as radar images rather than point clouds. Additionally, the partial point clouds in the dataset only contain $x$, $y$, and

---

The * denotes equal contribution. Correspondence to Xiangtai Li and Lu Qi.

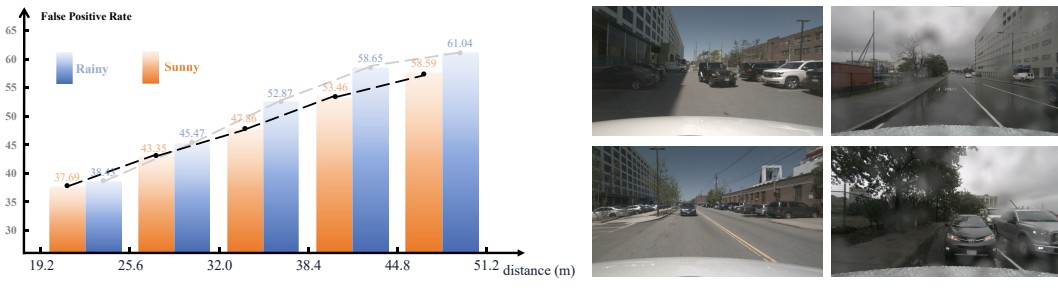

(a) Comparasion of radar point false positive rate on rainy and sunny days.

(b) Comparison of image visibility on rainy and sunny days.

Figure 1: **Illustration of radar and camera noise on sunny and rainy days.** Radar noise increases with distance from the radar sensor and is greater in rainy weather.

intensity dimensions, lacking many key characteristics of radar points, such as Radar Cross-Section (RCS) and Doppler speed.

To address the lack of radar-camera corruption datasets, we first simulate radar corruptions on the widely used and large-scale multi-modal dataset, nuScenes (Caesar et al., 2020). In particular, we focus on the graphic characteristics of corruption instead of the natural causes of corruption, exploring the optimal classification method for different noise patterns rather than being preoccupied with their causes. Our method can reduce overlaps between categories. For example, ground reflections or reflections caused by rainy or snowy weather, which are obviously different causes, may all result in radar echo disappearance. They fall into our first category of factors. As long as we can address the noise with the same pattern under all scenarios, the exact cause of the noise becomes less critical. This is because different interference conditions, such as multi-path effects and reflections, may change the distribution of radar point clouds in the same way, *i.e.*, point loss or the generation of false detection points. Additionally, building different types of noise distributions is more practical than using one specific noise source to enhance the model's robustness. Specifically, we consider four distinct noise patterns often occurring in radar sensor deployment and autonomous driving systems: **1) Key-point missing**, which manifests as the loss of radar points related to or unrelated to the target. **2) Spurious points**, which refers to the condition with false-positive radar points. **3) Point shifting**, representing radar points with deviations in the $x$, $y$, and $z$-axis due to interference, and **4) Non-positional disturbance**, referring to the situation where the position of radar points remains unchanged, but other characteristics such as RCS and Doppler speed deviate.

To address these issues, we introduce a robust 3D object detection framework named RobuRCDet, containing two critical designs for the robustness of both radar and camera signals. In this work, we propose a 3D Gaussian Expansion (3DGE) module to filter spatially inaccurate radar points through point expansion in the voxel field. We analyze the distribution pattern differences between the noisy and target point clouds. As shown in Figure 2, the noisy point is more randomly and sparsely distributed in space. We thus leverage the sparsity to sum all radar voxels and conduct normalization to enhance the dense part and reduce the sparse area according to the amplitude difference. Moreover, we design a Confidence-guided Multi-modal Cross-Attention (CMCA) module to enhance camera robustness by dynamically evaluating camera signal confidence. Since the confidence of the camera and radar signal varies significantly in different conditions, for instance, in adverse weather, the radar is much more robust than the camera. We exploit CMCA to learn reliable and accurate radar features from raw signals in proper conditions. Additionally, since the map is learned to have high camera signal confidence on high-quality camera images such as images on sunny days, it can preserve the original performance on clean data. With this module, we can preserve the original performance on clean data while ensuring robustness against noise. We conduct extensive experiments on original and augmented data and demonstrate the effectiveness and robustness of our method, especially under interfering conditions.

The main contributions of our works are as follows:

- To the best of our knowledge, we are the first to conduct systematic analysis on the robustness of radar-camera 3D object detection. We summarize the radar corruptions and establish a benchmark by simulating radar noises for robust 3D object detection evaluation.

- We propose a robust 3D object detector, RobuRCDet, to perform robust 3D object detection in various noise conditions. We design a 3D Gaussian Expansion module to highlight the key points and reduce the impacts from noisy points and a Confidence-guided Multi-modal Cross-Attention module to learn the robust multi-modal fusion.

- Extensive experimental results on nuScenes have shown the effectiveness of the proposed method. Our method achieves a 19.4% improvement in NDS and a 25.7% improvement in mAP under conditions of simultaneous radar signal interference and camera signal interference, compared to the baseline with only radar backbone and camera backbone.

## 2 RELATED WORK

**Camera-based 3D Object Detection.** The success of 2D object detection (Zhi et al., 2019; Xingyi et al., 2019), with the growing demand for 3D perception in fields like autonomous driving and robotics, prompts the development of 3D object detection technology. Early works (Li et al., 2023; Huang et al., 2021; Huang & Huang, 2022) are based on multi-view cameras which leverage multi-view information through cross-view interaction to improve 3D object detection performance. The multi-view 3D object detection methods can be briefly divided into two categories, *i.e.*, dense BEV-based and sparse query-based methods.

Numerous dense BEV-based methods adopt Lift-Splat-Shoot (LSS) (Philion & Fidler, 2020) to transform 2D features into BEV features, such as BEVDet (Huang et al., 2021). On the other hand, BEVDepth (Li et al., 2023) designs a trustworthy depth estimation module for better view transformation in the BEV space. For sparse query-based methods, PERT series (Liu et al., 2022; 2023a; Wang et al., 2023a;b) incorporate the position information of 3D coordinates into image features and integrate long-term temporal fusion.

While these methods achieve advanced 3D object detection performance, they overlook robustness, a key factor in real-world applications. Unlike millimeter-wave radar, camera images are prone to interference in darkness and bad weather, leading to poor detection performance. To this end, RobuRCDet incorporates the camera signal confidence map to effectively enhance network robustness along with radar modality under various conditions.

**Radar-Camera 3D Object Detection.** The camera sensor inherently lacks 3D depth information, limiting its 3D detection accuracy. To alleviate this issue, researchers propose incorporating the cost-effective radar sensor into the 3D detection framework. The radar sensor provides the 3D depth prior and additional Doppler velocity, compensating for the camera sensor's weaknesses.

Specifically, CenterFusion (Nabati & Qi, 2021) uses a key point detection network to obtain center points and then associates key points with the corresponding radar detection results in a pillar-based manner. After that, CRAFT (Kim et al., 2023a) further considers the spatial properties of radar and camera sensors and designs a proposal-level early fusion framework. RCBEV (Zhou et al., 2023) introduces the feature-level fusion in the BEV space for a unified feature representation. Meanwhile, RCM-Fusion (Kim et al., 2024) is proposed to combine radar and camera features at both the feature and instance levels, further improving the detection performance. More recently, CRN (Kim et al., 2023b) transforms PV image features to BEV with radar occupancy to compensate for the depth information in images. RCBEVDet (Lin et al., 2024) specifically customizes a feature extractor for radar and uses RCS as the object size prior. It further designs a Cross-Attention Multi-layer Fusion module for robust radar-camera feature alignment and fusion.

By contrast, focusing on the framework robustness, our RobuRCDet proposes a 3DGE module to decrease the impact of potential noisy points in the radar voxels.

**Robust 3D Object Detection.** Sensor noise is one of the most significant factors causing the decrease in detection performance during inference for 3D object detection. Several methods (Kong et al., 2023a;b; Xie et al., 2023; Kong et al., 2023c) attempt to benchmark the common corruptions in 3D perception tasks from different angles. For instance, RoboDepth (Kong et al., 2023c; Ren et al., 2022) sets up a benchmark to assess the robustness of monocular depth estimation in the presence of corruptions. On the other hand, RoboBEV (Xie et al., 2023) presents an extensive benchmark aimed at evaluating the robustness across four BEV perception tasks, *i.e.*, 3D object detection (Liu et al., 2023b; Liang et al., 2022) and semantic segmentation (Zhou & Krähenbühl, 2022). At the

| (a) Key-point Missing | (b) Spurious Points (random) | (c) Spurious Points (related) | (d) Point Shifting |
| --- | --- | --- | --- |

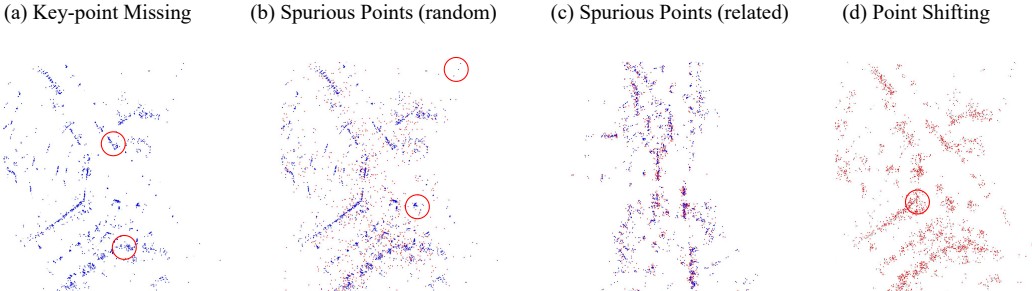

Figure 2: **Point cloud visualization of radar signals with noise.** The blue points refer to the ground truth radar points, and the red points represent noisy radar points in various conditions. Additionally, the light blue points in the key-point missing part denote the eliminated ground truth radar points.

same time, Robo3D (Kong et al., 2023b) evaluates the resilience of 3D detectors and segmenters when exposed to LiDAR-related corruptions. However, these benchmarks mostly focus on camera or lidar perceptions, but the radar corruptions are almost ignored.

Furthermore, the efforts (Jiang et al., 2024; Wu et al., 2024) are made to address the above-mentioned corruptions and achieve robust 3D object detection under noisy conditions. However, these methods only model partial radar noise types. Further, they only consider radar or LiDAR degradation scenarios and overlook camera failure cases. In contrast, our RoboRCDet addresses these issues and designs a module for the robust fusion of radar and camera features in the BEV view.

## 3 CORRUPTION TAXONOMY

Given the challenges of collecting real-world corruption data, we generate our training and validation dataset by synthesizing noise for radar and image signals. Since many existing methods focus on the robustness of image data, we focus more on exploring four noise types in radar signals: key-point missing, spurious points, point shifting, and non-positional disturbance.

### 3.1 RADAR SIGNAL

For clear illustration, a radar point $p \in \mathbb{R}^5$ is with coordinates $(x_p, y_p, z_p)$, Radar Cross-Section (RCS), and Doppler Speed $(v)$.

**Key-point missing.** The sparsity of radar point clouds poses a significant challenge to 3D detection. Interferences, such as reflections from lost radar beams, can worsen this issue, further increasing the sparsity of the point cloud. In our setting, we simulate key-point missing in two scenarios. Specifically, based on the existing clean radar points $p$, we either randomly or selectively remove points $R_k^\gamma(p)$ across the entire set or the region of the target object, which are represented by $\gamma = 0$ and $\gamma = 1$.

$$p_n = p - R_k^\gamma(p), k \in [1, M], \tag{1}$$

where $R$ denotes the random process to delete points from $p$. The $k$ is the number of points we should delete, ranging from 1 to $M$. The $M$ is set to be half of the number of $p$ or eight when $\gamma$ is 0 or 1. Finally, the kept final radar points are defined as $p_n$.

**Spurious points.** Due to intrinsic issues, noisy environments, or even artificial interference, spurious points can appear alongside the original radar point clouds. Similar to missing key points, spurious points are categorized into two types. The first type consists of noisy points superimposed on the original radar points, correlated with their positions, and randomly distributed around them. The second type contains completely random points originating from complex external environments and unrelated to the target point cloud.

$$p_n = p \cup p', \quad p' \sim N(\delta, \sigma), \quad \sigma \sim U(1, 50), \tag{2}$$

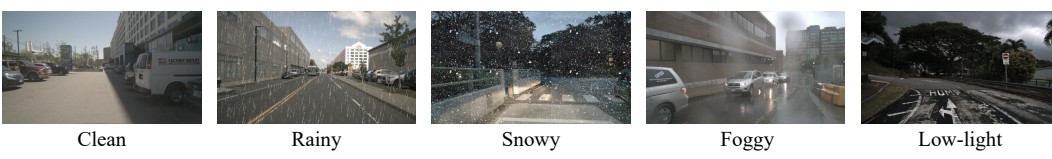

Figure 3: **Visualization of image signals under adverse weather conditions.**

$$\delta = \begin{cases} p(x_p, y_p, z_p, RCS, v), & \text{point related,} \\ R(x_p, y_p, z_p, RCS, v), & \text{random,} \end{cases} \qquad (3)$$

where $p'$ is the added noisy points. As shown in equation 2, the selection of $p'$ follows the normal distribution, where $\delta$ is defined in equation 3. Under the first circumstance, $\delta$ is set to $p$, while the $\delta$ is decided by a random process $R$ in the second situation.

**Point Shifting.** We refer to point shifting as the misalignment of 3D information where radar points deviate from their original locations. To simulate this corruption, we apply distortions directly to the radar points using a normal distribution $N(0, \sigma)$.

$$p_n = p + \Delta p, \qquad (4)$$

where $\Delta p \sim N(0, \sigma)$ and $\sigma \sim U(1, 50)$.

**Non-positional disturbance.** External interference can also affect the values of RCS and v, rather than introduce noise to the spatial coordinates. Although this scenario is less common than the previous three cases, we include it in our benchmark for completeness and refer to it as a non-positional disturbance.

As shown in equation 5, only the RCS and v dimensions are disturbed in the normal distribution manner, and the values of x, y, and z are consistent.

$$p_n = [x_p, \ y_p, \ z_p, \ RCS + \Delta RCS, \ v + \Delta v], \qquad (5)$$

where $\Delta RCS \sim N(0, \sigma)$, $\Delta v \sim N(0, \sigma)$, and $\sigma \sim U(1, 50)$. Visualization of the point cloud is shown in Figure 2.

## 3.2 IMAGE SIGNAL

We simulate images under adverse weather conditions, including rainy, snowy, foggy, and low light, according to the simulation methods in basic low-level tasks. The visual results of the synthetic degraded images are illustrated in Figure 3. **1) Adverse weather.** We leverage the composition method in (Han et al., 2022) to synthesize the images in various conditions. The rain, snow, and fog maps are provided, and we synthesize the normalized clean images. **2) Low light.** The low-light images are simulated through a classical gamma correction algorithm, where the gamma factor is randomly elected.

## 4 METHODOLOGY

Different from the concurrent works (Nabati & Qi, 2021; Zhou et al., 2023; Kim et al., 2023b), we propose RobuRCDet that focuses more on robust 3D object detection. As shown in Figure 4, our framework includes two separate branches for processing the image and radar point clouds and a fusion module. In the following subsections, we will overview the whole pipeline of RobuRCDet. Then, the proposed two modules will be elaborately described.

## 4.1 OVERVIEW OF ROBURCDET

As shown in Figure 4, we first pass the radar point cloud through a voxelization process. Then, 3DGE is applied to spread the RCS and Doppler speed dimensions to surrounding voxels according to a Gaussian distribution. The expanded radar voxels and the original radar voxels are then fed into

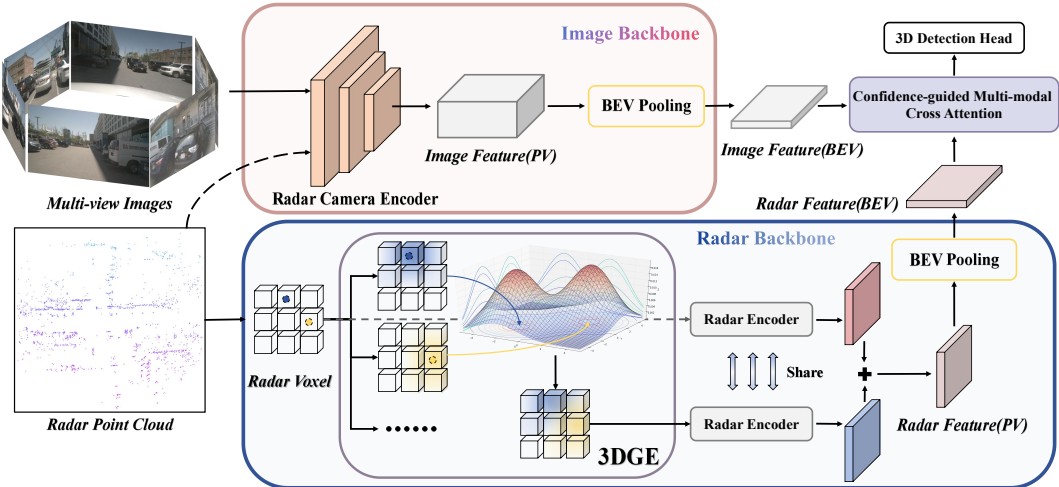

Figure 4: **Overall pipeline of the proposed RobuRCDet.** First, we extract the image features from multi-views and transform them into BEV space. Concurrently, 3DGE is employed on the radar voxels, and we put the original voxels and expanded voxels into the Radar Encoder to obtain radar features after summation. Finally, we fuse the image and radar features in the confidence-guided multi-modal cross-attention in the BEV space for 3D object detection.

a radar encoder with shared weights, resulting in the radar feature after feature summation. Next, both multi-view images and the radar point cloud are fed into the image backbone to extract image features guided by radar information. After transforming the radar and image features into the BEV space, CMCA is applied to fuse the features from both modalities. Finally, the fused features are used for 3D object detection tasks.

## 4.2 3D GAUSSIAN EXPANDING

To mitigate the impact of noisy radar points, we introduce the 3D Gaussian Expanding (3DGE) module to filter radar points in the voxel space. As illustrated in Figure 2, radar point distribution typically follows a pattern: points are dense within the target range, while false positive points are usually sparse. Here, our proposed 3DGE module leverages this semantic information in radar density, enhancing key points and suppressing false positives to handle extensive noise from radar corruption.

We note that even if interference causes key points to become sparser, radar points in the target region remain denser than in false positive areas, as shown in Figure 2. This is because

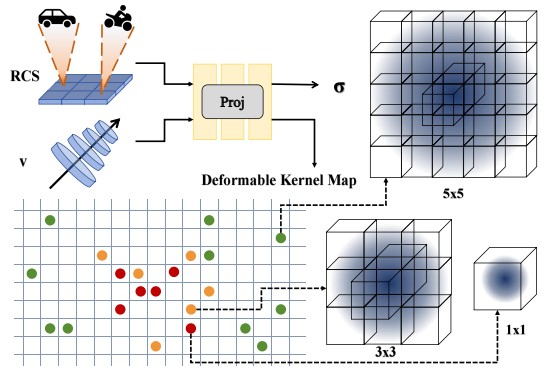

Figure 5: **Illustration of the 3D Gaussian Expanding.** It first utilizes a projector to learn the deformable kernel map and $\sigma$ from the RCS and velocity prior. After that, the two parameters are used to conduct expansion on each radar voxel.

false detection points exhibit strong randomness and are generally isolated or small clusters of scattered points. In contrast, target objects have a certain area, and multiple radar beams are typically reflected from this area. As long as the interference does not cause all beams from the same target object to vanish completely, the resulting point cloud is of high probability to be denser than that of the false detection interference. By applying 3DGE, these sparse key points can complement each other within the voxel, helping to restore the target area as much as possible and thereby maintaining the robustness of the radar branch.

Figure 5 illustrates the details of the proposed 3DGE module. First, we input the RCS and velocity information into a parameter encoder (Proj), generating the deformable kernel map and determining the variance of the Gaussian kernel. Next, we apply 3D Gaussian expansion to each radar point. Specifically, the RCS and velocity values are spread to the surrounding voxels of each radar point,

with the spreading range determined by the kernel size $\lambda_p$ provided by the deformable kernel map. For instance, if $\lambda_p$ is 3, the RCS and velocity are expanded into a $3 \times 3 \times 3$ region according to the normal distribution. To balance efficiency and accuracy, we restrict that $\lambda_p \in 1, 3, 5$. After expansion, the RCS and velocity values are summed within each voxel, followed by normalization to restore the values to their original range, as described by the following equation:

$$V_{radar}^{3DGE}(x, y, z, RCS, v) = \frac{V(RCS, v)}{2 \times \pi \times \sigma^2} \begin{cases} \exp^{\frac{(x-x_p)^2 + (y-y_p)^2}{2\sigma}}, & \begin{array}{l} |x - x_p| \in \lambda_p, \\ |y - y_p| \in \lambda_p, \end{array} \\ 0, & \text{otherwise,} \end{cases} \quad (6)$$

where $2 \times \pi \times \sigma^2$ is the coefficient of the Gaussian function, V represents the radar voxel, $x_p$ and $y_p$ are the x-coordinate and y-coordinate of the radar point, and we perform the expansion on the RCS and velocity dimensions. Next, normalization is applied in each expanded space to maintain the summation of the expanding kernel to 1 and prepare the voxel for feature extraction. Finally, the 3DGE result $V_{radar}^{3DGE}(x, y, z, RCS, v)$ combines the original radar voxel in a res-block manner.

## 4.3 CONFIDENCE-GUIDED MULTI-MODAL CROSS-ATTENTION

In this section, we introduce a Confidence-Guided Multi-Modal Cross-Attention (CMCA) module to address the challenges of low performance in degraded camera signals under adverse weather conditions and low-light scenarios. Radar demonstrates superior robustness compared to camera signals in many challenging scenarios, such as rainy and foggy days with very low visibility.

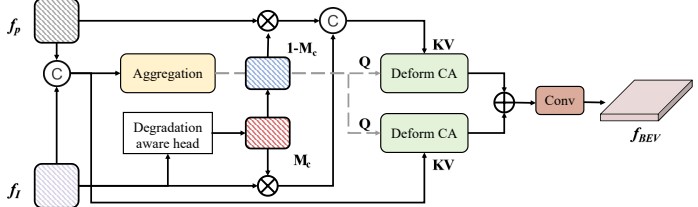

Figure 6: **Architecture of the Confidence-guided Multi-modal Cross-Attention module.** It considers the signal confidence of the camera and adaptively fuses image features and radar features to maintain robustness in various conditions.

As discussed in Section 1, in these situations, the confidence in radar signals is higher than that of camera signals. Therefore, using the fixed fusion method as in clear weather is unreasonable. We need an adaptive approach to adjust for both scenarios where the sensor confidences are similar (clear weather, *i.e.*, sunny) and where there is a significant confidence disparity (low visibility conditions, *i.e.*, foggy, rainy, snowy).

To ensure the inference speed of the model and reduce training time costs, we do not apply specific evaluation or constraint mechanisms, such as prompts, loss functions, or image quality assessment methods, to the CMCA module. Instead, we utilize the existing nighttime and rainy scenes in the nuScenes training dataset to guide the degradation-aware head in dynamically learning optimal performance strategies. As shown in Figure 6, we leverage the image feature $f_I$ as the input of the degradation aware head to generate the camera signal confidence map $M_c$ as follows:

$$M_c = \text{Softmax}(\text{MLP}(f_I)), \quad (7)$$

which is decided by the degradation level of the camera signal. Next, we multiply $f_I$ by $M_c$ to get $f_I^c$ and multiply $f_p$ by $1 - M_c$ to get $f_p^c$, allowing the information from $M_c$ to guide the adaptive fusion of radar and image features thoroughly. Next, we concatenate $f_I$ and $f_p$, then conduct aggregation to obtain $f_A$ as Q while obtaining $f_{mm}$ as:

$$\begin{aligned} f_A &= W(\text{Concat}(LN(f_I), LN(f_p))), \\ f_{mm} &= \text{Concat}(LN(M_c \times f_I), LN((1 - M_c) \times f_p)), \end{aligned} \quad (8)$$

where $W$ indicates the linear projection, $LN$ refers to the LayerNorm and Concat refers to the concatenation process.

This adaptive adjustment mechanism calculates the confidence of the signals for each scene. In adverse weather conditions, the map will generate low confidence, guiding the fuser to integrate radar features while deeply minimizing interference from image signals. In addition, since the detection accuracy of radar camera-based methods primarily comes from the camera, we rely more on the camera signal when it has high confidence, with the radar signal serving as adaptive support.

Table 1: **3D Object Detection on nuScenes `val` set.** 'C' and 'R' represent camera and radar, respectively. Some results are borrowed from RCBEVDet (Lin et al., 2024).

| Methods | Input | Backbone | Image Size | NDS↑ | mAP↑ | mATE↓ | mASE↓ | mAOE↓ | mAVE↓ | mAAE↓ |
|---|---|---|---|---|---|---|---|---|---|---|
| CenterFusion | C+R | DLA34 | 448×800 | 45.3 | 33.2 | 0.649 | 0.263 | 0.535 | 0.540 | 0.142 |
| CRAFT | C+R | DLA34 | 448×800 | 51.7 | 41.1 | 0.494 | 0.276 | 0.454 | 0.486 | 0.176 |
| RCBEV4d | C+R | Swin-T | 256×704 | 49.7 | 38.1 | 0.526 | 0.272 | 0.445 | 0.465 | 0.185 |
| CRN | C+R | R18 | 256×704 | 54.2 | 44.9 | 0.518 | 0.283 | 0.552 | 0.279 | 0.180 |
| RCBEVDet | C+R | R18 | 256×704 | 54.8 | 42.9 | 0.502 | 0.291 | 0.432 | 0.210 | 0.178 |
| RobuRCDet | C+R | R18 | 256×704 | **55.0** | **45.5** | 0.516 | 0.287 | 0.521 | 0.281 | 0.184 |
| BEVDet | C | R50 | 256×704 | 39.2 | 31.2 | 0.691 | 0.272 | 0.523 | 0.909 | 0.247 |
| BEVDepth | C | R50 | 256×704 | 47.5 | 35.1 | 0.639 | 0.267 | 0.479 | 0.428 | 0.198 |
| SOLOFusion | C | R50 | 256×704 | 53.4 | 42.7 | 0.567 | 0.274 | 0.411 | 0.252 | 0.188 |
| StreamPETR | C | R50 | 256×704 | 54.0 | 43.2 | 0.581 | 0.272 | 0.413 | 0.295 | 0.195 |
| CRN | C+R | R50 | 256×704 | 56.0 | 49.0 | 0.487 | 0.277 | 0.542 | 0.344 | 0.197 |
| RCBEVDet | C+R | R50 | 256×704 | **56.8** | 45.3 | 0.486 | 0.285 | 0.404 | 0.220 | 0.192 |
| RobuRCDet | C+R | R50 | 256×704 | 56.7 | **51.2** | **0.481** | **0.273** | 0.499 | 0.317 | 0.193 |

Finally, we apply the multi-scale deformable cross-attention (Deform CA) to generate the BEV feature. For the original feature fusion, we concatenate $f_I$ and $f_p$ to be the key and Value, while for the confidence-aware feature fusion, we concatenate $f_I^c$ and $f_p^c$ to be the key and Value. Then, the summation and convolution are conducted on the outputs of Deform CA modules to form the BEV feature $f_{BEV}$. The overall equation can be depicted as:

$$f_{BEV} = \text{Conv}(\text{Deform CA}(f_A, \text{Concat}(f_I, f_p)) + \text{Deform CA}(f_A, f_{mm})), \qquad (9)$$

where $f_A$ refers to the features obtained from the sparse aggregation module. Through CMCA, we invite adaptability into the radar feature and image feature fusion process, which maintains the robustness of camera signals in multiple conditions.

## 5 EXPERIMENTS

### 5.1 EXPERIMENTAL SETUP

**Datasets and Evaluation Metrics.**

We train and evaluate our method on the widely used benchmark nuScenes (Caesar et al., 2020). We add the simulated radar and camera corruption to form the noisy dataset on nuScenes. We use the official metrics for the 3D object detection task, including NDS and mAP.

**Implementation Details.** For the camera stream, we adopt the image

Table 2: **Corruption results on nuScenes `val` set.** C1 to C3 represent the Spurious Points, Non-positional Disturbance, and Key-point Missing, respectively. In addition, for the first two corruptions, the level refers to $\sigma$ while the level of C3 refers to the number of missing beams.

| Corruption | | CRN | | RCBEVDet | | RobuRCDet | |
|---|---|---|---|---|---|---|---|
| Type | level | NDS↑ | mAP↑ | NDS↑ | mAP↑ | NDS↑ | mAP↑ |
| C1 | 3 | 44.6 | 39.0 | 47.4 | 39.3 | 47.0 | 41.2 |
| | 5 | 39.2 | 36.0 | 44.2 | 38.9 | 44.6 | 40.5 |
| C2 | 3 | 37.3 | 35.4 | 41.7 | 39.6 | 42.2 | 40.6 |
| | 5 | 34.8 | 32.1 | 36.5 | 32.7 | 37.4 | 35.1 |
| C3 | 10 | 50.1 | 41.9 | 52.4 | 42.7 | 52.7 | 43.8 |
| | 14 | 48.7 | 39.6 | 51.0 | 40.5 | 50.4 | 41.9 |
| Rain | - | 42.1 | 31.2 | 45.6 | 32.9 | 45.9 | 33.6 |
| Fog | - | 46.8 | 37.7 | 51.9 | 43.1 | 51.3 | 43.6 |
| Snow | - | 41.6 | 30.1 | 44.1 | 31.7 | 44.7 | 32.8 |
| Night | - | 38.2 | 31.4 | 42.1 | 35.0 | 42.6 | 39.0 |

encoder in CRN (Kim et al., 2023b) with several modifications; That is, we add a confidence map projector to extract confidence map from image feature in the BEV space. For the radar, we accumulate eight previous radar sweeps and use normalized RCS and Doppler speed as input features following GRIF (Kim et al., 2020) Net and CRN (Kim et al., 2023b). Our model is trained for 24 epochs with AdamW (Loshchilov, 2017) optimizer. We apply image and BEV data augmentation (Li et al., 2023) to prevent overfitting. In addition, we randomly drop sweeps and points for radar input following (Leng et al., 2023).

### 5.2 MAIN RESULTS

**Clean Results.** We compare RobuRCDet with previous state-of-the-art 3D detection methods on the `val` set, as shown in Table 1. The results show that RobuRCDet achieves competitive perfor-

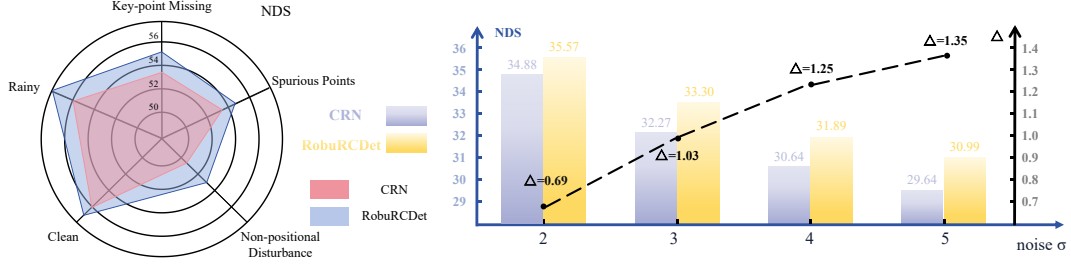

(a) Comparasion of CRN and RobuRCDet on five degradation types.  (b) Comparasion of CRN and RobuRCDet on Point Shifting with various σ.

Figure 7: **Validation on various corruptions.** All the metrics are tested on models with ResNet-50 backbone. In figure (a), 'Clean' and 'Rainy' are tested on the real-world dataset, while the radar corruptions are augmented with $sigma = 1$. In figure (b), the $\Delta$ refers to the RobuRCDet's performance growth over CRN.

Table 3: **Ablation of the main components of RobuRCDet.** We progressively integrate components into BEVDepth (Li et al., 2023) and the PointPillar encoder, forming RobuRCDet. Additionally, IB and RB refer to the Image Backbone and Radar Backbone, respectively.

| IB | RB | 3DGE | CMCA | Normal Condition | | | | Corruption Condition | | | |
|---|---|---|---|---|---|---|---|---|---|---|---|
| | | | | NDS↑ | mAP↑ | mATE↓ | mAP (Car)↑ | NDS↑ | mAP↑ | mATE↓ | mAP (Car)↑ |
| ✓ | | | | 43.9 | 33.2 | 0.716 | 50.4 | - | - | - | - |
| ✓ | ✓ | | | 54.3 | 42.4 | 0.536 | 68.4 | 28.5 | 23.9 | 0.709 | 39.5 |
| ✓ | ✓ | ✓ | | 54.9↑0.6 | **46.1**↑3.7 | 0.523↓0.013 | **71.5**↑3.1 | 33.6 ↑5.1 | 29.4↑5.5 | 0.677↓0.032 | 47.3↑7.8 |
| ✓ | ✓ | | ✓ | **55.2**↑0.9 | 45.8↑3.4 | 0.531↓0.005 | 70.7↑2.3 | 33.1↑4.6 | 28.6↑4.7 | 0.681↓0.028 | 46.7↑7.2 |
| ✓ | ✓ | ✓ | ✓ | 55.0↑0.7 | 45.5↑3.1 | **0.516**↓0.020 | 70.7↑2.3 | 34.1↑5.6 | 30.07↑6.14 | 0.635↓0.074 | 48.7↑9.2 |

mance compared to previous methods. Specifically, with ResNet-18 as the image backbone, RobuR-CDet increases mAP by 2.6 and 0.6 compared to RCBEVDet and CRN, respectively. Notably, for the ResNet-50 backbone, our method surpasses CRN and RCBEVDet on mAP by 4.5% and 13.0%, respectively, showing the effectiveness of RobuRCDet on detection tasks.

**Corruption Results.** In Table 2, we illustrate the performance of RobuRCDet and another two state-of-the-art models CRN and RCBEVDet on various augmented corruptions. Specifically, we achieve 44.7 NDS and 32.8 mAP on snowy test sets, surpassing CRN by 3.1 NDS and 2.7 mAP.

In addition, as shown in Figure 7, we illustrate the comparison of CRN and our proposed method on real-scenario data and each radar corruption mentioned in Section 3.1. Figure 7 (a) shows that our method performs more robustly in all scenes than CRN. Furthermore, in Figure 7 (b), CRN has a 5.24 NDS performance drop from $\sigma = 2$ to $\sigma = 5$, while RobuRCDet merely decreases 4.58 NDS, which improves robustness by 12.6%.

## 5.3 ABLATION STUDIES

We perform ablation studies on the nuScenes `val` set to evaluate the effectiveness of each configuration of RobuRCDet. The baseline model uses RobuRCDet with a ResNet-18 backbone, an image size of 256×704, and a BEV size of 128×128.

**Main Components.** As shown in Table 3, we integrated 3DGE and CMCA into the baseline to enhance the detector's robustness. All results are obtained from models trained on the clean dataset, and we evaluate them on both the clean validation set (referred to as the Normal Condition) and the synthesized noisy validation set (referred to as the Corruption Condition). Under Normal Conditions, it is notable that 3DGE and CMCA improve NDS by 0.6 and 0.9, respectively, with 3DGE achieving a 3.1 increase in mAP for cars.

Furthermore, under the Corruption Condition, 3DGE improves NDS by 17.67% and mAP by 22.82%, which indicates that 3DGE can achieve favorable performance under strong radar interference. Additionally, the higher performance increase over that under Normal Conditions demonstrates the robustness of the proposed components.

Table 4: **Ablation of 3DGE.** Uniform refers to the kernel size remaining fixed during expansion, while Ada indicates an adaptively sized kernel map. Additionally, wxyz signifies that the input to the parameter encoder includes RCS and $v$ along with $x$, $y$, and $z$.

| Method | Clean | | | | Key-point Missing | | | |
|---|---|---|---|---|---|---|---|---|
| | NDS↑ | mAP↑ | mAOE↓ | mAP (Car)↑ | NDS↑ | mAP↑ | mAOE↓ | mAP↑ |
| Baseline | 53.7 | 44.2 | 0.563 | 70.1 | 53.6 | 44.0 | 0.563 | 70.1 |
| +uniform 3DGE | 52.9 | 44.0 | 0.551 | 70.1 | 51.4 | 43.1 | 0.562 | 68.6 |
| +Ada 3DGE (wxyz) | 50.7 | 42.7 | 0.607 | 69.5 | 49.5 | 41.9 | 0.622 | 68.5 |
| +Ada 3DGE (ours) | **54.8** | **45.5** | **0.523** | **70.7** | **54.7** | **45.3** | **0.522** | **70.5** |

Table 5: **Validation on real word interference.** We selected data from the nuScenes dataset under challenging lighting and weather conditions to validate the effectiveness of RobuRCDet in real-world scenarios.

| Method | input | Night | | | Rainy | | |
|---|---|---|---|---|---|---|---|
| | | NDS↑ | mAP↑ | mAP (Car)↑ | NDS↑ | mAP↑ | mAP (Car)↑ |
| CRN (Kim et al., 2023b) | C+R | 33.3 | 25.2 | 73.0 | 56.1 | 47.3 | 76.3 |
| CRN+CMCA (Kim et al., 2023b) | C+R | 33.6↑0.3 | 25.9↑0.7 | 73.1↑0.1 | 57.5↑1.4 | 48.0↑0.7 | 76.7↑0.4 |
| RCBEVDet (Lin et al., 2024) | C+R | 34.4 | 25.3 | **73.8** | **59.4** | 47.1 | 76.9 |
| RobuRCDet (ours) | C+R | **35.5** | **28.2** | 73.4 | 58.4 | **49.2** | **77.8** |

**3DGE module.** In Table 4, we conduct the ablation experiments of 3DGE design, especially for the part of the deformable kernel map. Notably, utilizing a uniform kernel map or inviting position information $(x, y, z)$ to learn the kernel map and $\sigma$ is not beneficial to the detection performance. This is because the key to determining whether a radar point is a false positive lies not in its position but rather in its RCS and Doppler speed. In addition, uniform kernel size may blur the boundaries of the target objects, which are close in distance, resulting in detection difficulty and performance loss. Specifically, our adaptive 3DGE increases the NDS and mAP by 1.1, 1.3, and 1.9, 1.5 compared to the baseline and uniform 3DGE on the clean dataset. In addition, 3DGE decreases the mAOE by 7.3% under the Key-point Missing conditions compared to the baseline model. To this end, adaptive 3DGE is the most effective design compared to the uniform 3DGE and adaptive 3DGE with position information. Furthermore, adaptive 3DGE achieves higher performance than the baseline in both interference and non-interference environments.

## 5.4 ANALYSIS OF ROBUSTNESS

Table 5 illustrates the performance of RobuRCDet and previous state-of-the-art methods under real-world challenging lighting and weather conditions, *i.e.*, rainy conditions and night conditions. In rainy conditions, we achieve 1.8 NDS, and 1.4 mAP over CRN while at night time RobuRCDet surpasses RCBEVDet by 3.20% in NDS and 11.46% in mAP. In addition, we replace the MDCA module in CRN with our CMCA module. CRN achieves 56.1 NDS and 47.3 mAP in rainy scenarios and the incorporation of CMCA yields 1.4 NDS and 0.7 mAP improvement, which demonstrates the effectiveness and transferable characteristics of CMCA.

Furthermore, to enable RobuRCDet to achieve better performance under challenging conditions and to demonstrate the robustness and effectiveness of the proposed method, we also conducted training and testing on the noisy dataset. The results can be found in the supplementary materials.

## 6 CONCLUSION

We introduce RobuRCDet, a radar-camera fusion method designed to enhance the robustness of 3D object detection. Our approach addresses the challenges of strong interference and suboptimal performance in diverse perception conditions by designing two key modules: 3DGE and CMCA. Experimental results demonstrate that RobuRCDet outperforms previous state-of-the-art radar-camera 3D object detection methods in challenging conditions.

**Acknowledgments.** This work was supported by National Key R&D Program of China (Grant No. 2022ZD0160305).

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

## A OVERVIEW

This supplementary material provides additional details of architecture, and qualitative and quantitative experimental results. We describe implementation details for experiments in the main paper (Section B). We further provide additional experimental results on noisy training sets (Section C) and qualitative results (Section D).

## B IMPLEMENTATION DETAILS

This section provides some experimental settings and network details in the main paper.

First, for the network detail and hyper-parameters, we employ SECONDFPN (Yan et al., 2018) to concatenate output feature maps at stride 16 and let the output depth bins of the depth distribution network to be 112 with a depth range of [2.0, 58.0]m and bin size to be 0.5m. For the CMCA module, we use the multi-scale deformable attention implementation from MMCV (Contributors, 2018) and set the number of attention heads to 8 and sampling points to 2. We set the radar point range as [-51.2, 51.2]m and make the BEV feature map $128 \times 128$.

Second, for the noisy training set, we set the ratio of clean images to noisy images in the training set to 8:2. For the noisy images, we randomly synthesize one of the four types of proposed radar corruptions, with the intensity of the noise also being random. Additionally, we synthesize harsh weather or low-light conditions on the images corresponding to the timestamps with noise.

In addition, the corruption levels of Spurious Points, Point Shifting, and Non-positional Disturbance are determined by $\sigma$, while the number of missing beams determines the noise level of Key-point Missing. Furthermore, weather degradation is also classified into levels; for example, rain and fog can be divided into light or heavy, whereas snow does not have a classification. Additionally, the low-light level at night is determined by the gamma coefficient, with a mild low-light coefficient set to 1.0-2.0 and a heavy set to 2.0-3.0.

## C ANALYSIS ON THE NOISY TRAINING SET.

Table 6 illustrates the results of CRN and RobuRCDet which are trained on the noisy dataset and tested on each corruption. ResNet-18 is used as the backbone. In this table, C0 denotes the clean testing set and it is notable that RobuRCDet achieves the same performance (54.4 NDS and 44.9 mAP) as the model trained on the clean dataset and surpasses CRN by 1.6 NDS and 1.4 mAP. Furthermore, C1 to C4 represent Spurious Point, Non-positional Disturbance, Key-point Missing, and Point Shifting.

According to Table 6 and the comparison in Table 2, performance is improved when processing noisy radar point clouds after training with the corruption training set. Additionally, to ensure fairness in the experiments, we train and test both the RobuRCDet and CRN methods on the disturbed dataset and compared their performance. It is clear that our method still demonstrates better robustness than CRN with 2.4 NDS and 1.9 mAP improvement in C1 with $\sigma = 10$.

## D ADDITIONAL VISUAL RESULTS.

In this section, we present more synthesized noisy images. Notably, to better simulate real scenarios, we ensure that the degradation types and levels for multiple cameras at the same timestamp are the same. Figure 8 and Figure 9 showcase the synthesized images of rainy days, snowy days, and nighttime mentioned in the main paper.

## E SIMULATION OF THE EFFECT OF 3DGE ON THREE TYPES OF NOISE.

In this section, we included simulations of the effects of 3DGE on various types of data, which are shown in Figure 10. These simulation results visually demonstrate the functioning and effectiveness of 3DGE. For instance, although the patterns of the three types of noise differ, the surrounding noise points consistently appear as deep blue, indicating that, after processing, their impact on the

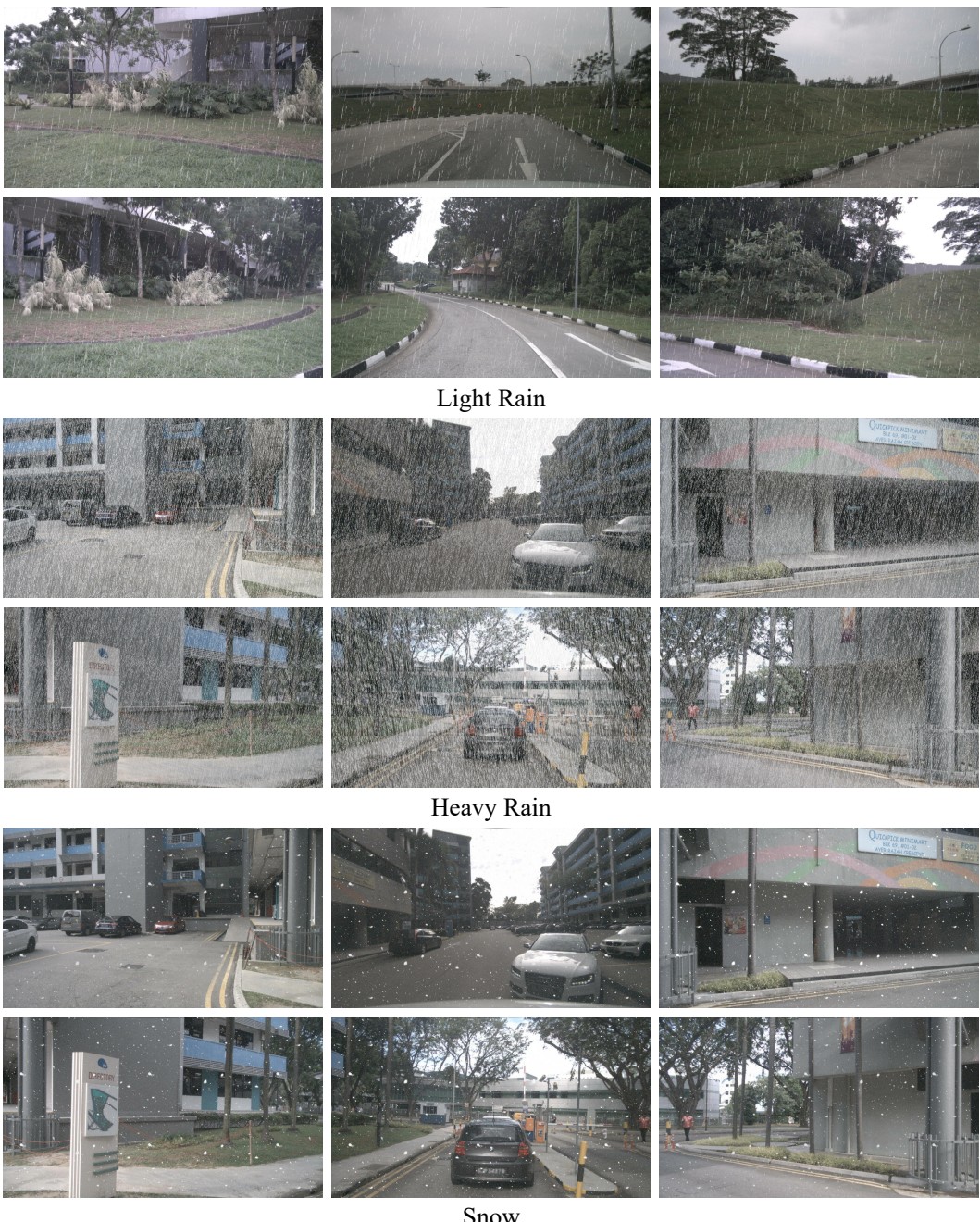

Light Rain

Heavy Rain

Snow

Figure 8: **Visualization of synthesized challenging weather images.** Two levels of rainy images and a set of snowy images are displayed.

Table 6: **Validation of models trained with noisy training sets.** We augment the image data and radar data to form the training set. denotes that the model is retrained.

| Corruption | | CRN* | | | | RobuRCDet | | | |
|---|---|---|---|---|---|---|---|---|---|
| Type | level | NDS↑ | mAP↑ | mATE↓ | mAP (Car) ↑ | NDS↑ | mAP↑ | mATE↓ | mAP (Car)↑ |
| C0 | - | 52.8 | 43.5 | 0.550 | 69.6 | 54.4 | 44.9 | 0.517 | 70.9 |
| C1 | 1 | 52.1 | 42.9 | 0.553 | 69.1 | 54.0 | 44.7 | 0.524 | 70.6 |
| | 10 | 51.2 | 42.4 | 0.560 | 68.4 | 53.6 | 44.3 | 0.532 | 70.1 |
| C2 | 1 | 51.6 | 42.6 | 0.557 | 69.2 | 53.4 | 44.1 | 0.539 | 70.1 |
| | 10 | 50.5 | 41.1 | 0.568 | 68.1 | 52.6 | 42.9 | 0.547 | 69.7 |
| C3 | 8 | 52.3 | 43.0 | 0.551 | 69.2 | 54.1 | 44.6 | 0.537 | 70.2 |
| | 10 | 52.0 | 42.3 | 0.569 | 68.8 | 53.8 | 44.0 | 0.642 | 69.9 |
| C4 | 1 | 41.6 | 35.2 | 0.668 | 55.4 | 45.1 | 36.9 | 0.637 | 56.8 |
| | 10 | 33.4 | 28.0 | 0.750 | 41.5 | 36.7 | 31.2 | 0.699 | 47.0 |

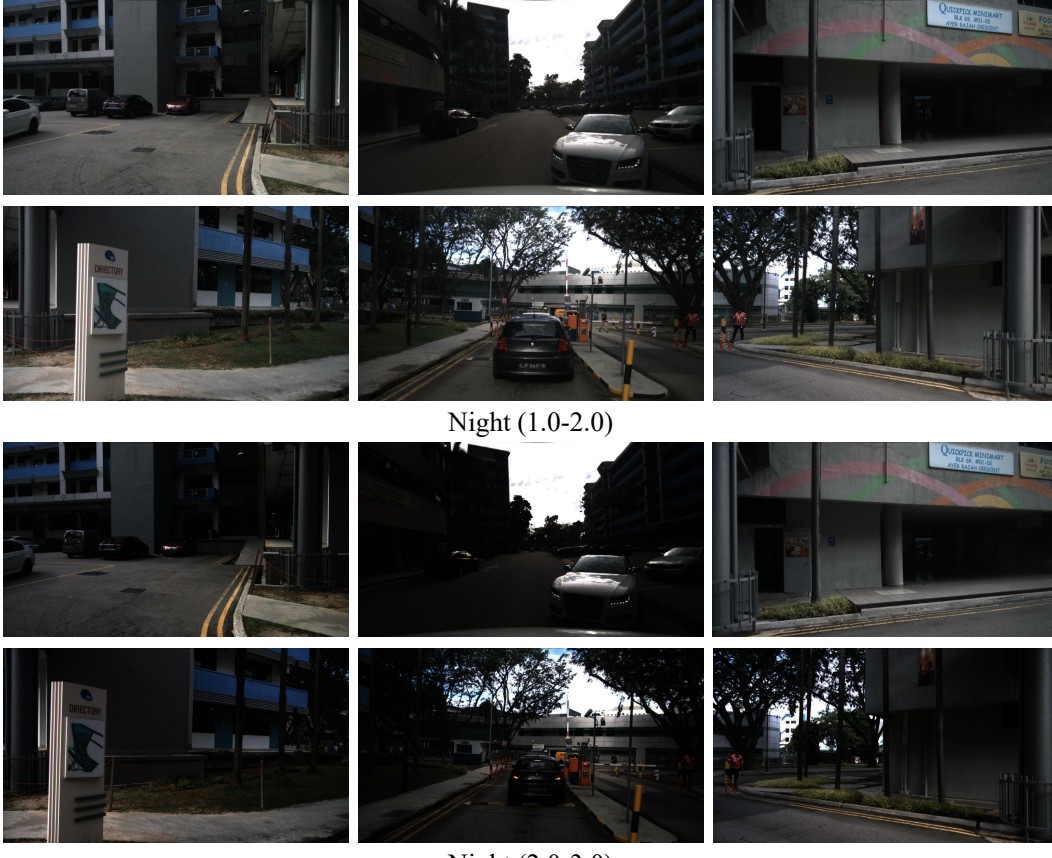

Night (1.0-2.0)

Night (2.0-3.0)

Figure 9: **Visualization of synthesized challenging light conditions.** Two levels of low-light images are displayed.

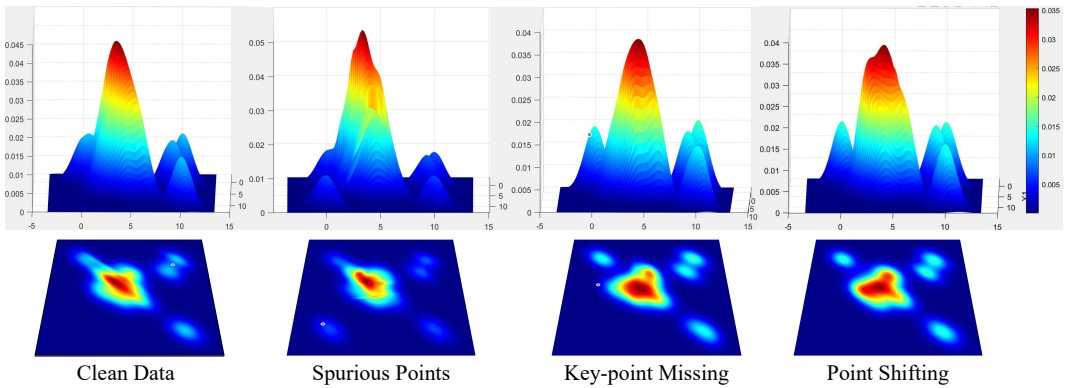

Figure 10: **Simulation of the effect of 3DGE on three types of noise.**

recognition target is minimal. Furthermore, even though the shapes of the heatmaps around the target vary after processing, the deep red regions, representing the peak positions of the targets, remain entirely consistent. Notably, the spurious points appearing around the target region can even contribute to strengthening the target area and diminishing the influence of surrounding points.

