# OpenReview forum: "RobuRCDet: Enhancing Robustness of Radar-Camera Fusion in Bird's Eye View for 3D Object Detection"
_ICLR.cc/2025/Conference — ICLR 2025 Poster_

### Official Review · Reviewer_YH6m · 2024-11-03

**Soundness:** 2
**Presentation:** 3
**Contribution:** 2
**Rating:** 5
**Confidence:** 5

**Summary:**

The paper introduces RobuRCDET, a novel approach to effectively fuse radar and camera features for 3D object detection. The core idea is to suppress false radar points by predicting Gaussian kernel variance. The approach demonstrates promising results on the nuScenes validation set.

**Strengths:**

+ The paper is well-written and easy to comprehend.
+ The core idea of suppressing false radar points by predicting Gaussian kernel variance is nice and leverages the information bottleneck principle.
+ The proposed method of weighting camera and radar streams leads to robust feature representation.
+ The approach demonstrates promising results on the nuScenes validation set.

**Weaknesses:**

- The idea of predicting Gaussian variance in 3DGE module like decomposition and re-combining is one of the ways to denoise radar points. Another way to denoise is using self-attention blocks [A]. How does the method work when you replace the 3DGE module by multiple self-attention blocks.

- The claim of "Extensive Experiments" in the abstract is exaggerated. It would be beneficial to quantitatively include results from the nuScenes leaderboard, particularly comparing against a strong camera baseline like SparseBEV with 640x1600 resolution.

- The experiments are conducted with super small backbones (ResNet18 and ResNet50). It would be insightful to quantitatively evaluate the approach on higher resolutions (512x1508 and 640x1600) to assess its performance.

- The paper focuses on mid-level feature fusion. A quantitative comparative analysis with end-level fusion, as employed in RADIANT [B], would provide valuable insights. To further strengthen the argument for mid-level fusion, incorporating the radar association module from RADIANT should be considered.


References:
- [A] Unveiling the Hidden Structure of Self-Attention via Kernel Principal Component Analysis, Teo et al, NeurIPS 2024
- [B] RADIANT: Radar Image Association Network for Radar-Camera Object Detection, Long et al, AAAI 2023

**Questions:**

Please see the weakness. I will need nuscenes leaderboard results on the 640x1600 resolution and comparison against SparseBEV 640x1600 resolution.

---

> ### Author Response · Authors · 2024-11-22
> **Response to Reviewer YH6m**
>
> **Q1. The performance of replacing the 3DGE module with multiple self-attention blocks.**
>
> We referred to the attention method mentioned in [1] and replaced 3DGE to conduct the experiment, with the results shown in the table below. It is noticeable that our method with 3DGE surpasses self-attention by  1.3 NDS on clean data and 5.2 NDS on Sporious Points noise, verifying the effectiveness of the proposed method.
>
> | Method        | Clean | Key-point Missing | Spurious Points | Point Shifting | Non-positional Disturbance |
> |---------------|-------|-------------------|------------------|----------------|----------------------------|
> | Self-Attention | 55.4  | 50.8              | 41.8             | 28.9           | 39.6                       |
> | 3DGE          | 56.7  | 52.7              | 47.0             | 33.3           | 42.2                       |
>
> ---
>
> **Q2. Experimental setup for larger backbone networks and higher resolutions.**
>
> Our method focuses on model robustness, aiming to maintain strong robustness while minimizing performance drop on clean datasets, rather than pursuing higher accuracy alone. Additionally, Robust 3D detection is typically deployed in vehicle-side applications, where lightweight models are generally preferred, such as ResNet-18 or ResNet-50 with a 704x256 resolution. Therefore, we primarily considered the application of lightweight models, ResNet-18 and ResNet-50, in the main paper.
>
> Moreover, to further demonstrate the effectiveness of RobuRCDet, we provide the experimental results with a larger backbone (ResNet-101) and a higher resolution (1408x512) according to the reviewer's suggestion. As shown in the table below, our method outperforms CRN and SparseBEV by 0.9 NDS, indicating the effectiveness of the proposed method under various settings.
>
>
> | Method   | Input | Backbone | Image Size | NDS↑  | mAP↑  | mATE↓ |
> |----------|-------|----------|------------|-------|-------|-------|
> | SparseBEV | C     | R101     | 1408x512   | 59.2  | 50.1  | 0.562 |
> | CRN      | C+R   | R101     | 1408x512   | 59.2  | 52.5  | 0.460 |
> | Ours     | C+R   | R101     | 1408x512   | 60.1  | 53.4  | 0.452 |
>
> ---
>
> **Q3. Supplementing with the radar association module from RADIANT.**
>
> RobuRCDet belongs to the same category as BEVFusion [2], which directly fuses multi-modal features into a single BEV feature and predicts with only one branch. In contrast, RADIANT processes image and radar features separately, resulting in outputs from two branches that can utilize an association module. Thus, the radar association module cannot be applied to our method directly. In fact, our method employs attention mechanisms to implicitly perform the association in the BEV space.
>
> [1] Teo et al. Unveiling the Hidden Structure of Self-Attention via Kernel Principal Component Analysis. NeurIPS, 2024.
>
> [2] Liu Z, Tang H, Amini A, et al. Bevfusion: Multi-task multi-sensor fusion with unified bird's-eye view representation. ICRA, 2023.

---

> > ### Author Response · Authors · 2024-11-24
> > **Follow-Up on Rebuttal Discussion**
> >
> > Dear Reviewer YH6m,
> >
> > We deeply appreciate your valuable feedback during the first round of review and the thoughtful discussion that has significantly helped us refine our work, Since the discussion phase ends on Nov 26, we would like to know whether we have addressed all the issues, and we would greatly welcome any additional feedback or suggestions you may have.
> >
> > Thank you again for your devotion to the review! If all the concerns have been successfully addressed, please consider raising the scores after this discussion phase.
> >
> > Best regards,
> >
> > Paper2908 Authors

---

> > ### Comment · Reviewer_YH6m · 2024-12-01
> > **Response to Rebuttal.**
> >
> > Thank you for the rebuttal. I read response from other reviewers' as well. However, I maintain my original score as the authors have not addressed the following concerns:
> >
> > - The paper does not demonstrate improved performance over SparseBEV at **640x1600** resolution on the **nuScenes leaderboard**, even though SparseBEV has released their leaderboard model at 640x1600 resolution. This raises doubts about the effectiveness of RobuRCDet at higher resolutions, limiting its practical utility in cloud-deployments. It's **absolutely crucial** for 3D detection methods published in top-tier conferences like ICLR, NeurIPS, ICCV or CVPR to include nuScenes leaderboard results. An example of this is CRN (in their Table 2).
> >
> > - Additionally, while AP shows a 3.3-point improvement over SparseBEV at 1408x512 resolution, the NDS gain is only 0.9 points. This suggests a significant decrease in mAVE, mAAE, mAOE, and mASE when using RobuRCDet (NDS is 50% AP and 50% TP metrics). While I am OK with decrease in mAAE, mAOE and mASE metrics, I do not understand why does mAVE of RobuRCDet become worse compared with SparseBEV after including radar, especially when the radar provides radial velocity in RobuRCDet.

---

> > > ### Author Response · Authors · 2024-12-02
> > > **Follow-Up on Rebuttal Discussion**
> > >
> > > **Q1. Experiment for resolution 640*1600.**
> > >
> > >
> > > The following table compares the metrics of RobuRCDet and SparseBEV on V2-99. To ensure fairness in the comparison, our experimental setup is fully aligned with SparseBEV: we use a total of 8 frames and do not include future frames. As shown in the table below, our method achieves an improvement of 0.8 NDS and 2.8 mAP compared to SparseBEV.This demonstrates that RobuRCDet is also effective with a large backbone.
> > >
> > >
> > > | Method     | Input  | Backbone | Image Size | NDS↑ | mAP↑ | mATE↓  |
> > > |------------|--------|----------|------------|-------|-------|--------|
> > > | SparseBEV  | C      | V2-99    | 1600x640   | 63.6  | 55.0  | 0.485  |
> > > | Ours       | C+R    | V2-99    | 1600x640   | 63.8  | 57.1  | 0.407  |
> > >
> > >
> > > **Q2. Question about mAVE, mAOE,and mAAE decrease.**
> > >
> > > The meanings of mAVE, mAOE and mAAE are as follows:
> > >
> > > **mAOE**:
> > > Average Orientation Error. The Average Orientation Error (AOE) is the smallest yaw angle difference between predicted and ground truth values. (All category angle deviations are within 360°, except for the "obstacle" category, where angle deviations are within 180°.)
> > >
> > > **mAVE**:
> > > Average Velocity Error. The Average Velocity Error (AVE) is the L2 norm of the 2D velocity difference (m/s).
> > >
> > > **mAAE**:
> > > Average Attribute Error. The Average Attribute Error (AAE) is defined as \(1 - \text{accuracy}\), where accuracy (\(acc\)) is the classification accuracy of the attributes.
> > >
> > > We use CRN as our baseline. Therefore, some fluctuations in certain metrics are due to architectural differences, since CRN inherently performs worse than SparseBEV in metrics like mAVE (0.093↑), mAOE (0.155↑), and mAAE (0.011↑) on ResNet50. As for SparseBEV, it differs from CRN, RCBEVDet, and our method in terms of architecture. SparseBEV is based on a sparse transformer head, while all of our methods are based on BEV and our method outperforms CRN at all aspects.
> > > Additionally, to deal with the poor performance on mAOE, mAAE, and mAVE, we will design 3DGE as a transferable module and attempt to enhance the performance of other baselines.

---

> ### Author Response · Authors · 2024-11-27
> **Please let us know whether all issues are addressed**
>
> Dear reviewer,
>
> Thanks for the comments and review. We have provided more explanations and answers to your questions. Since the deadline for discussion is near the end, please let us know whether we have answered all the questions. Please also consider raising the score after all issues are addressed.
>
> If you have more questions, please raise them and we will reply ASAP.
>
> Thanks,
>
> Authors

---

> > ### Author Response · Authors · 2024-12-01
> > **Follow-Up on Rebuttal Discussion**
> >
> > Dear Reviewer YH6m,
> >
> > Thank you once again for your insightful feedback. With the deadline approaching on December 2, we would greatly appreciate the opportunity to clarify any remaining concerns or answer any questions you may have.
> >
> > If all issues have been addressed to your satisfaction, we kindly ask you to consider revising the scores accordingly after this discussion phase. We look forward to your continued feedback and hope to resolve any lingering doubts as efficiently as possible.
> >
> > Thank you again for your time and dedication to this review!
> >
> > Best，
> >
> > Authors

---

> ### Author Response · Authors · 2024-12-04
> **Please let us know whether all issues are addressed**
>
> Dear reviewer,
>
> Thanks for the comments and review. We have provided more explanations and answers to your questions. Since the deadline for discussion is near the end, please let us know whether we have answered all the questions.
>
> If you have more questions, please raise them and we will reply ASAP.
>
> Thanks,
>
> Authors

---

### Official Review · Reviewer_B47b · 2024-11-04

**Soundness:** 2
**Presentation:** 2
**Contribution:** 2
**Rating:** 6
**Confidence:** 4

**Summary:**

The paper proposes a BEV space object detector using camera and radar data that uses a 3D Gaussian to mitigate radar noise and adaptively fuses camera and radar features based on camera feature quality. The 3D Gaussian Edge module learns to spread the RCS and velocity values to surrounding voxels and the Confidence-Guided Multi-modal cross attention module learns to adaptively fuse radar and camera features by learning to detect degradation of the image features. Training and evaluation is done using simulated nuScenes dataset and shows improvement over CRN and RCBEVDet.

**Strengths:**

The main strength of the paper is that it considers different types of sensor degradation and proposes a radar data expansion and camera-radar fusion approach to mitigate those degradations.

**Weaknesses:**

The proposed 3DGE seems to be densifying the radar data based on the reasoning in lines 306-316. How do the other noise types, i.e., spurious points, point shifting and non-positional disturbances get mitigated? Furthermore, spurious points may get worse due to being spread across multiple voxels, and there’s no discussion on how non-positional disturbances are addressed through 3DGE. Furthermore, the ablation experiments only include keypoint noise and missing the other 3 types of radar noises as mentioned in the paper.

There’s no discussion on the training process. How does the model learn M_c from the data? Adverse data are rare and what prevents the network from always choosing image features?

All the results are based on simulated data therefore the conclusions from the results would depend a lot on the fidelity of the simulation. There isn’t much discussion on how the radar noise simulations were done. As for the image signal, the performance will be limited by [Han et al. 2022] for adverse weather. For low-light, cameras have signal dependent noise characteristics which cannot be modeled by random gamma factor.

**Questions:**

Line 69: What is meant by “focus on the corruption graphic characteristics instead of the natural causes of the corruption”? If the noise distribution of the corruption doesn’t match the noise characteristics of the radar then the resulting model doesn’t add much benefit in practice.

Figure 2: how were the noise parameters of the plots determined? How were the ground truth points determined in the captured data, i.e., they can already have the 4 types of radar noise.

How is the camera signal confidence reliably learned in practice with imbalance data?

The need for the learned 3D Gaussian Expanding component is unclear especially given that the set of lambda_p is small, how does the model perform without learning the sigma and simply performing deformable convolution on a 5x5x5 grid?

How well does the proposed approach work on real adverse weather and noisy data?

---

> ### Author Response · Authors · 2024-11-22
> **Response to Reviewer B47b Part(1/3)**
>
> We thank reviewer B47b for acknowledging the contribution of our paper and providing thoughtful comments.
> We would like to address the raised concerns as follows:
>
> **Q1. The mitigation mechanism of 3DGE for Spurious Points, Point shifting, and Non-positional Disturbance noise.**
>
> The design of 3DGE is ingenious. It is not solely achieved through densification but rather fully leverages the characteristic that radar points on targets are denser than noise scatter points. The mechanism for different types of noise is as follows:
>
> (1) **Key-point Missing**: Even when some of the key points are missing (assuming the missing points are uniformly distributed or do not completely obscure a target), the target's point cloud remains denser than other regions. In this case, applying 3DGE supplements the densification at the target's location. For highly dense target point clouds, the kernel size tends to remain 1, while for sparse point clouds, the kernel size may increase to 3.
>
> (2) **Spurious Points**: Targets naturally become denser due to 3DGE. In dense regions, the kernel size tends to stay at 1. During the overlapping process, the inherent density of the target point cloud, combined with the noise points, can cause the peak values in the target area to become significantly higher than those in noise scatter regions. This intensity difference helps us better identify the target.
>
> (3) **Point Shifting**: Similar to the mechanisms in dealing with Spurious Points, 3DGE highlights dense regions, blurring the effects of point shifts.
>
> (4) **Non-positional Disturbance**: The distribution of noise in non-positional is Gaussian distribution with a mean of 0. 3DGE can use a larger kernel size to average the non-positional noise, which can reduce the noise deviations close to the mean value of 0.
>
> Additionally, based on the results shown in Table 2, our method also demonstrates certain advantages over other methods in handling Non-positional Disturbance noise.
>
> | Corruption Type             | Level | CRN NDS↑ | CRN mAP↑ | RCBEVDet NDS↑ | RCBEVDet mAP↑ | RobuRCDet NDS↑ | RobuRCDet mAP↑ |
> |-----------------------------|-------|----------|----------|---------------|---------------|----------------|----------------|
> | **Non-positional Disturbance** | 3     | 37.3     | 35.4     | 41.7          | 39.6          | 42.2           | 40.6           |
> |  **Non-positional Disturbance**   | 5     | 34.8     | 32.1     | 36.5          | 32.7          | 37.4           | 35.1           |
>
> ---
>
>
> **Q2. The ablation study lacks the inclusion of the other three types of noise.**
>
> In Table 2, we present the individual results for various simulated noise types and weather conditions. In addition, the results of noisy training are provided in the supplementary materials. Due to space constraints of the main paper, we only included the results of single noise types in the ablation study section.
>
> Furthermore, we included simulations of the effects of 3DGE on various types of data in the supplementary materials. These simulation results visually demonstrate the functioning and effectiveness of 3DGE. For instance, although the patterns of the three types of noise differ, the surrounding noise points consistently appear as deep blue, indicating that, after processing, their impact on the recognition target is minimal. Furthermore, even though the shapes of the heatmaps around the target vary after processing, the deep red regions, representing the peak positions of the targets, remain generally consistent. Notably, the spurious points appearing around the target region can even contribute to strengthening the target area and diminishing the influence of surrounding points.
>
> ---

---

> > ### Author Response · Authors · 2024-11-22
> > **Response to Reviewer B47b Part(2/3)**
> >
> > **Q3. The learning method of M_c.**
> >
> > To ensure the inference speed of the model and reduce training time costs, we do not apply specific evaluation or constraint mechanisms, such as prompts, loss functions, or image quality assessment methods, to the CMCA module. Additionally, the labeling of adverse weather conditions is typically performed by humans, who may assign lower confidence to rainy images. However, this approach may not yield the best performance. According to the table below, the camera confidence remains high on rainy days.
> >
> > Instead, we utilize the existing nighttime and rainy scenes in the nuScenes training dataset, as well as synthesized adverse weather scenarios at specific ratios, to guide the degradation-aware head in dynamically learning optimal performance strategies. In the table, the M_c of nighttime images is noticeably low, while the mean value for rainy days is slightly higher than that of the entire validation set. This is partly because the validation set contains a small proportion of nighttime images, limiting their overall impact due to their low ratio. Moreover, most rainy-day images in the nuScenes dataset exhibit relatively mild degradation, with targets remaining clearly visible. This results in higher camera confidence scores.
> >
> >
> > | Data Split      | val           |Rainy|Night|
> > |----------------|----------------|-----|-----|
> > | Mean Value of M_c| 0.64|0.65|0.32|
> >
> >
> > Additionally, we have added details about the learning process of the M_c parameter in the main paper: "To ensure the inference speed of the model and reduce training time costs, we do not apply specific evaluation or constraint mechanisms, such as prompts, loss functions, or image quality assessment methods, to the CMCA module. Instead, we utilize the existing nighttime and rainy scenes in the nuScenes training dataset, as well as synthesized adverse weather scenarios at specific ratios, to guide the degradation-aware head in dynamically learning optimal performance strategies."
> >
> > ---
> >
> > **Q4. The performance of RobuRCDet in handling real-world noise.**
> >
> > For the real-world camera noise, we present the results under real rainy and nighttime conditions in Table 5.
> >
> > | Method       | Night NDS↑ | Night mAP↑ | Night mAP(Car)↑ | Rainy NDS↑ | Rainy mAP↑ | Rainy mAP(Car)↑ |
> > |--------------|------------|------------|-----------------|------------|------------|-----------------|
> > | CRN          | 33.3       | 25.2       | 73.0            | 56.1       | 47.3       | 76.3            |
> > | CRN+CMCA     | 33.6       | 25.9       | 73.1            | 57.5       | 48.0       | 76.7            |
> > | RCBEVDet     | 34.4       | 25.3       | 73.8            | 59.4       | 47.1       | 76.9            |
> > | Ours         | 35.5       | 28.2       | 73.4            | 58.4       | 49.2       | 77.8            |
> >
> > For the real-world radar noise,  we conducted tests across different distance ranges (from 0 to 51.2m in radius) and compared it with CRN. We used NDS as the evaluation metric to demonstrate the effectiveness of RobuRCDet. The results show that although both methods' performance declines with increasing distance, the drop in performance for RobuRCDet (0.6 NDS) is noticeably smaller than that of CRN (1.6 NDS). This further validates the effectiveness of our method in handling real radar noise.
> >
> > | Method | [0,12.8) | [12.8,25.6) | [25.6,51.2) | Average |
> > |---------|----------|-------------|-------------|---------|
> > | CRN     | 56.9     | 56.2        | 55.3        | 56.0    |
> > | Ours    | 57.1     | 56.9        | 56.5        | 56.7    |
> >
> > ---
> > **Q5. Line 69: What is meant by “focus on the corruption graphic characteristics instead of the natural causes of the corruption”? If the noise distribution of the corruption doesn’t match the noise characteristics of the radar then the resulting model doesn’t add much benefit in practice.**
> >
> > The statement means that we aim to explore the optimal classification method for different noise patterns rather than being preoccupied with their causes.
> >
> > Our method can reduce overlaps between categories. For example, ground reflections or reflections caused by rainy or snowy weather, which are obviously different causes, may all result in radar echo disappearance. They fall into our first category of factors. As long as we can address the noise with the same pattern under all scenarios, the exact cause of the noise becomes less critical.
> >
> > We have included this explanation in the final version of the paper.

---

> > > ### Author Response · Authors · 2024-11-22
> > > **Response to Reviewer B47b Part(3/3)**
> > >
> > > **Q6. Figure 2: how were the noise parameters of the plots determined?**
> > >
> > > The noise parameters are empirical and partly based on our experimental results. Additionally, we determined a range of degradation parameters by referencing the degradation levels in RADIATE [2].
> > >
> > > ---
> > >
> > > **Q7. How were the ground truth points determined in the captured data, i.e., they can already have the 4 types of radar noise.**
> > >
> > > Since current technology cannot ensure that radar operates completely noise-free. For instance, as shown in Figure 1 of the main paper, noise points often appear in long-distance regions.
> > >
> > > Due to the complexity and unpredictability of the real-world environment, it is difficult to classify these noise points into a specific category of the proposed noise types. This especially highlights the necessity and innovation of 3DGE, which can handle all four types of noise simultaneously. Real-world noise is often a mixture of these four types, and the ability to address them collectively ensures better applicability in real scenarios. For example, the results in Table 1 were obtained on the nuScenes dataset, where the radar data inherently contains unavoidable noise—specifically, the false detection rate illustrated in the introduction. As shown in Table 1, our method achieves excellent performance metrics even on this naturally noisy dataset.
> > >
> > > Additionally, the design of the four noise patterns was partly inspired by well-established LiDAR noise models, with modifications made to account for differences between LiDAR and radar.
> > >
> > > ---
> > >
> > > **Q8. The need for the learned 3D Gaussian Expanding component is unclear especially given that the set of lambda_p is small, how does the model perform without learning the sigma and simply performing deformable convolution on a 5x5x5 grid?**
> > >
> > > We note that applying deformable convolution directly to voxels is not feasible, as the voxel data format is \(n, M, c)\, where n represents the number of non-empty voxels, M is the maximum number of points per voxel (fixed at 8 in our experiments), and c represents the 5 dimensions of radar points: \(x, y, z, RCS, v\). This format does not meet the requirements for deformable convolution.
> > >
> > > Furthermore, we carry out experiments to answer this question. Our detailed solution is to keep the kernel size to 3x3x3 in the main paper as the table below. However, the performance is bad and we are not sure whether this experiment meets your needs. We will continue to explore this part.
> > >
> > > | Method         | Clean Data NDS↑ | Clean Data mAP↑ | Clean Data mAOE↓ | Clean Data mAP(Car)↑ |
> > > |----------------|-----------------|-----------------|-------------------|----------------------|
> > > | uniform 3DGE   | 52.9            | 44.0            | 0.551             | 70.1                 |
> > > | 3DGE           | 54.8            | 45.5            | 0.523             | 70.7                 |
> > >
> > >
> > > [1] Tian X, Jiang T, Yun L, et al. Occ3d: A large-scale 3d occupancy prediction benchmark for autonomous driving. NeurIPS, 2024.
> > >
> > > [2] Sheeny M, De Pellegrin E, Mukherjee S, et al. Radiate: A radar dataset for automotive perception in bad weather. ICRA, 2021.

---

> > > > ### Author Response · Authors · 2024-11-24
> > > > **Follow-Up on Rebuttal Discussion**
> > > >
> > > > Dear Reviewer B47b,
> > > >
> > > > We deeply appreciate your valuable feedback during the first round of review and the thoughtful discussion that has significantly helped us refine our work, Since the discussion phase ends on Nov 26, we would like to know whether we have addressed all the issues, and we would greatly welcome any additional feedback or suggestions you may have.
> > > >
> > > > Thank you again for your devotion to the review! If all the concerns have been successfully addressed, please consider raising the scores after this discussion phase.
> > > >
> > > > Best regards,
> > > >
> > > > Paper2908 Authors

---

> > > ### Author Response · Authors · 2024-11-27
> > > **Please let us know whether all issues are addressed**
> > >
> > > Dear reviewer,
> > >
> > > Thanks for the comments and review. We have provided more explanations and answers to your questions. Since the deadline for discussion is near the end, please let us know whether we have answered all the questions. Please also consider raising the score after all issues are addressed.
> > >
> > > If you have more questions, please raise them and we will reply ASAP.
> > >
> > > Thanks,
> > >
> > > Authors

---

> ### Author Response · Authors · 2024-12-01
> **Follow-Up on Rebuttal Discussion**
>
> Dear Reviewer B47b,
>
> Thank you once again for your insightful feedback. With the deadline approaching on December 2, we would greatly appreciate the opportunity to clarify any remaining concerns or answer any questions you may have.
>
> If all issues have been addressed to your satisfaction, we kindly ask you to consider revising the scores accordingly after this discussion phase. We look forward to your continued feedback and hope to resolve any lingering doubts as efficiently as possible.
>
> Thank you again for your time and dedication to this review!
>
> Best,
>
> Authors

---

> ### Author Response · Authors · 2024-12-04
> **Please let us know whether all issues are addressed**
>
> Dear reviewer,
>
> Thanks for the comments and review. We have provided more explanations and answers to your questions. Since the deadline for discussion is near the end, please let us know whether we have answered all the questions. Please also consider raising the score after all issues are addressed.
>
> If you have more questions, please raise them and we will reply ASAP.
>
> Thanks,
>
> Authors

---

### Official Review · Reviewer_MsLQ · 2024-11-04

**Soundness:** 3
**Presentation:** 2
**Contribution:** 3
**Rating:** 6
**Confidence:** 3

**Summary:**

The paper addresses the problem of robustness in radar-camera fusion techniques for 3D object detection. The authors point out that adverse weather conditions, poor lighting, and sensor noise often cause existing methods to fail specifically because of the "flat" fusion approach that is usually used. The paper introduces RobuRCDet to overcome the shortcomings of existing approaches by using confidence-based fusion.
Key contributions:
Analysis of common noise types affecting radar data in real-world scenarios (key-point missing, spurious points, point shifting, non-positional disturbance) and create a benchmark by simulating noise patterns for evaluating robustness.
A model with 2 key contributions: 3D Gaussian Expansion (3DGE) for filtering out noisy radar points based on the sparsity distribution. And a Confidence-guided Multimodal Cross Attention (CMCA) for dynamically and reliably fusing the radar and camera features based on the confidence in the camera signal.
Ablation studies to corroborate the effectiveness of the added contributions on noisy radar and camera signals.

**Strengths:**

The paper is tackling a relevant problem hindering the reliability of machine learning approaches for sensor fusion in challenging scenarios. The related work is comprehensive and detailed. The approach presented is simple yet effective and builds on top of proven concepts. Although "confidence-based fusion" is not a new concept in itself and has been used for a long time in classical fusion approaches (e.g. Kalman Filters), the approach presented by the authors seems to be effective while not overly complicated. The authors combine multiple tried and proven ideas to achieve their results.

**Weaknesses:**

While the idea is presented clearly, there seem to be some missing definitions of parameters used in equations that are possible to infer but could be clearer. The diagrams, although readable, they can be more detailed to reflect the equations and information in the text. One of the methods mentioned in Table 1 is not cited (StreamPETR).

**Questions:**

In the voxelization and kernel generation approaches there are unclarities or unanswered questions:
what voxel size is used and how does it affect the quality of the detection?
if there are multiple targets in the same voxel, does that affect the computation of the 3DGE? equation 6 indicates otherwise but this does not make sense since a voxel with more targets should have more influence than a voxel with a single target.
While nuScenes radar data includes the "z" value of the radar, it is unclear how accurate that value is and in reality, most of the radars on market are 2.5D and not 3D, aka they do not have a proper way to measure the elevation (no resolution in elevation) and thus the cartesian z value. How would the results of the 3D detection change if no "z" values were used?

---

> ### Author Response · Authors · 2024-11-22
> **Response to Reviewer MsLQ**
>
> We thank reviewer MsLQ for acknowledging the contribution of our paper and providing thoughtful comments.
> We would like to address the raised concerns as follows:
>
> **Q1. Missing Definitions of Parameters in Equations.**
>
> In the updated version, we have incorporated your suggestions by adding parameter definitions. We add the definition of **x_p** and **y_p** as the x-coordinate and y-coordinate of the radar point in lines 333-334.
>
> ---
>
> **Q2. The Improvement of Diagrams and Presentation.**
>
> In our revised version, we have carefully updated the diagrams and presentation to ensure they provide more detailed and intuitive visual representations of the corresponding content.
>
> ---
>
> **Q3. The Missing Citation in Table1.**
>
>
> We have cited the StreamPETR [1] in lines 124-126 of the manuscript, please see the revised version.
>
> ---
>
> **Q4. What voxel size is used and how does it affect the quality of the detection?**
>
> In the BEV space, the voxel size for the x and y axes are both 0.2m. Regarding its impact on detection, empirically, smaller voxel sizes generally lead to higher accuracy, while the computational cost increases exponentially. For example, the following table shows the detection results of a popular 3D detector, CenterPoint [2], with different voxel sizes.
>
> | Voxel Size          | NDS↑ | mAP↑ |
> |---------------------|-------|------|
> | (0.075, 0.075, 0.2) | 67.3  | 60.3 |
> | (0.1, 0.1, 0.2)     | 65.3  | 58.0 |
>
> ---
>
> **Q5. The impact of the number of targets within voxels on the computation of 3DGE.**
>
> The impact of the number of targets within voxels on the computation of 3DGE is minimal. This is because 3DGE fundamentally processes each point contained within a voxel. Multiple targets result in multiple intensity peaks, and 3DGE is designed to be deformable to handle densely populated point cloud regions. In such dense areas, the network learns to adopt smaller kernel sizes, preserving the inherent features of the point cloud. This can be verified by the added simulation result in the supplementary material.
>
> Additionally, the voxel format processed by the Voxelization function in our code by **mmcv** library is represented as (n, M, C), where **n** denotes the number of non-empty voxels. In our experiments, **n** generally equals the total number of points in the point cloud, making target overlap unlikely.
>
> ---
>
> **Q6. The impact of a single voxel containing multiple targets on Equation 6.**
>
> Since the voxel size (0.2m) in our experiments is small, the density of voxels typically causes a single target to be distributed across multiple voxels rather than multiple targets being contained within a single voxel. Even if a voxel contains multiple points, we set the maximum number of points per voxel to 8 in our experiments. Under normal circumstances, 8 points are insufficient to fully represent a single target. Therefore, the impact of a single voxel containing multiple targets is small.
>
> ---
>
> **Q7. Ablation results for the missing z-dimension condition.**
>
> We conducted experiments by removing the z-dimension using RobuRCDet. The results showed that although the performance was slightly worse than in the 3D case, the decrease in metrics was limited, showing a drop from 55.0 NDS to 54.7 NDS. This demonstrates the robustness of our method and highlights its practical applicability for commercial millimeter-wave radar systems.
>
> | Method     | NDS↑ | mAP↑ | mATE↓ | mASE↓ | mAOE↓ | mAVE↓ | mAAE↓ |
> |------------|-------|------|-------|-------|-------|-------|-------|
> | without z  | 54.7  | 45.1 | 0.527 | 0.283 | 0.531 | 0.267 | 0.185 |
> | with z     | 55.0  | 45.5 | 0.516 | 0.287 | 0.521 | 0.281 | 0.184 |
>
>
> [1] Wang S, Liu Y, Wang T, et al. Exploring object-centric temporal modeling for efficient multi-view 3d object detection. ICCV, 2023.
>
> [2] Yin T, Zhou X, Krahenbuhl P. Center-based 3d object detection and tracking. CVPR, 2021.

---

> ### Author Response · Authors · 2024-11-24
> **Follow-Up on Rebuttal Discussion**
>
> Dear Reviewer MsLQ,
>
> We deeply appreciate your valuable feedback during the first round of review and the thoughtful discussion that has significantly helped us refine our work, Since the discussion phase ends on Nov 26, we would like to know whether we have addressed all the issues, and we would greatly welcome any additional feedback or suggestions you may have.
>
> Thank you again for your devotion to the review! If all the concerns have been successfully addressed, please consider raising the scores after this discussion phase.
>
> Best regards,
>
> Paper2908 Authors

---

> ### Author Response · Authors · 2024-11-27
> **Please let us know whether all issues are addressed**
>
> Dear reviewer,
>
> Thanks for the comments and review. We have provided more explanations and answers to your questions. Since the deadline for discussion is near the end, please let us know whether we have answered all the questions. Please also consider raising the score after all issues are addressed.
>
> If you have more questions, please raise them and we will reply ASAP.
>
> Thanks,
>
> Authors

---

### Official Review · Reviewer_uUpc · 2024-11-08

**Soundness:** 3
**Presentation:** 2
**Contribution:** 3
**Rating:** 8
**Confidence:** 3

**Summary:**

This paper conducts a systematic analysis of radar-camera detection robustness under five types of noise and proposes RobuRCDet, a robust object detection model in bird’s-eye view (BEV). To address radar point inaccuracies, including position, Radar Cross-Section (RCS), and velocity, this work introduces a 3D Gaussian Expansion (3DGE) module. This module uses RCS and velocity priors to create a deformable kernel map, adjusting kernel size and value distribution. Additionally, this paper proposes a weather-adaptive fusion module that dynamically merges radar and camera features based on camera signal confidence. Experiments show the effectiveness of the proposed RobuRCDet.

**Strengths:**

(1)The figures in this paper are well-crafted.
(2)The proposed 3D Gaussian Expanding method is both novel and effective, as demonstrated by the experiments.

**Weaknesses:**

1) The CMCA module seems to be a standard method, how does the degradation-aware head assess the confidence of the camera and radar features?
2) Although Pepper is designed as a robust fusion method between radar and camera, its performance is not much stronger than that of RCBEVDet in tab.2, particularly regarding the NDS.

**Questions:**

please see the weakness section

---

> ### Author Response · Authors · 2024-11-22
> **Response to Reviewer uUpc**
>
> We thank reviewer uUpc for acknowledging the contribution of our paper and providing thoughtful comments.
> We would like to address the raised concerns as follows:
>
> **Q1. The Method of Degradation-aware Head in CMCA.**
>
> To ensure the inference speed of the model and reduce training time costs, we do not apply specific evaluation or constraint mechanisms, such as prompts, loss functions, or image quality assessment methods, to the CMCA module. Additionally, the labeling of adverse weather conditions is typically performed by humans, who may assign lower confidence to rainy images. However, this approach may not yield the best performance. According to the table below, the camera confidence remains high on rainy days.
>
> Instead, we utilize the existing nighttime and rainy scenes in the nuScenes training dataset, as well as synthesized adverse weather scenarios at specific ratios, to guide the degradation-aware head in dynamically learning optimal performance strategies. In the table, the M_c of nighttime images is noticeably low, while the mean value for rainy days is slightly higher than that of the entire validation set. This is partly because the validation set contains a small proportion of nighttime images, causing a slight impact due to the mean of M_c . Moreover, most rainy-day images in the nuScenes dataset exhibit relatively mild degradation, with targets remaining clearly visible. This results in higher camera confidence scores.
>
>
> | Data Split      | val           |Rainy|Night|
> |----------------|----------------|-----|-----|
> | **Mean Value of M_c**| 0.64|0.65|0.32|
>
> ---
>
> **Q2. A slight performance disadvantage compared to RCBEVDet.**
>
> We will continue to update the high-resolution version (1600x900) and a version with a more suitable number of radar sweeps of RobuRCDet in the future to achieve higher accuracy and robustness.

---

> > ### Author Response · Authors · 2024-11-24
> > **Follow-Up on Rebuttal Discussion**
> >
> > Dear Reviewer uUpc,
> >
> > We deeply appreciate your valuable feedback during the first round of review and the thoughtful discussion that has significantly helped us refine our work, Since the discussion phase ends on Nov 26, we would like to know whether we have addressed all the issues, and we would greatly welcome any additional feedback or suggestions you may have.
> >
> > Thank you again for your devotion to the review!
> >
> > Best regards,
> >
> > Paper2908 Authors

---

### Author Response · Authors · 2024-11-22
**Summary**

We thank all reviewers uUpc, MsLQ, B47b and YH6m for their positive feedback:
* The proposed 3D Gaussian Expanding method is both novel and effective (uUpc, MsLQ, B47b and YH6m).
* The work is tackling a reliability problem of machine learning approaches (MsLQ, B47b).
* Well writen and figure well crafted (uUpc, YH6m). The related work is comprehensive and detailed (MsLQ).
* The The approach demonstrates promising results on the nuScenes validation set (YH6m).

In the following, we address the raised issues of each reviewer.

---

### Meta-Review · Area_Chair_hhts · 2024-12-21

**Metareview:**

This paper proposes a camera-radar fusion method for 3D object detection, demonstrating better robustness compared to pure image-based methods in adverse weather conditions. Three reviewers provided positive evaluations, while one reviewer maintained a negative stance. In their response, the authors effectively addressed this reviewer's concerns about high-resolution experimental performance and clearly explained the reasons for performance differences with the SparseBEV method. The reviewer did not provide comments on the author's feedback. After reading the discussion and other reviews, the AC believes the authors have adequately addressed this reviewer's concerns. Therefore, considering all the reviews, the final recommendation is accept.

**Additional Comments On Reviewer Discussion:**

This paper was reviewed by four reviewers and received initial scores of 8, 6, 5, and 5. After the rebuttal period, Reviewer B47b changed the score from 5 to 6. Other reviewers kept their scores unchanged without further comments. After reviewing all the comments and author feedback, the AC believes that the authors have adequately addressed this reviewer's concerns. Therefore, the final recommendation is accept.

---

### Decision · Program_Chairs · 2025-01-22

Accept (Poster)